# Impact of quarantine due to COVID-19 pandemic on health and lifestyle conditions in older adults from Centro American countries

**Neyda Ma. Mendoza-Ruvalcaba[1], Raúl Gutiérrez-Herrera[2], Cecilia López[3], Heike Hesse[4], Marcio Soto-Añari [5], Miguel Ramos-Henderson[6], Juan-Carlos Cárdenas-Valverde[7], Loida Camargo[8], Nicole Caldichoury[9], Jorge Herrera-Pino[10], José Calizaya-López[11], Cesar Castellanos[12], Claudia García[13], María F. Porto[14], Norman López[14]***

**1** Departamento de Ciencias de la Salud-Enfermedad como Proceso Individual, Universidad de Guadalajara CUTONALA, Tonala, Mexico, **2** Facultad de Medicina, Universidad Autónoma de Nuevo León, San Nicolás de los Garza, Nueva León, México, **3** Asociación Grupo Ermita Alzheimer de Guatemala, Ciudad de Guatemala, Guatemala, **4** Observatorio COVID-19, Universidad Tecnológica Centroamericana, Tegucigalpa, Honduras, **5** Laboratorio de Neurociencia, Universidad Católica San Pablo, Arequipa, Perú, **6** Centro de Investigación e Innovación en Gerontología Aplicada (CIGAP), Universidad Santo Tomás, Antofagasta, Chile, **7** Universidad Nacional de Huancavelica, Huancavelica, Perú, **8** Escuela de Medicina, Universidad del Sinú, Cartagena de Indias, Colombia, **9** Departamento de Ciencias Sociales, Universidad de Los Lagos, Osorno, Chile, **10** College of Medicine, Florida International University, Miami, Florida, United States of America, **11** Universidad Nacional de San Agustín de Arequipa, Arequipa, Perú, **12** Instituto Dominicano para el Estudio de la Salud Integral y la Psicología Aplicada (IDESIP), Santo Domingo, República Dominicana, **13** Departamento de Neuropsicología, Facultad de Ciencias Sociales, Universidad del Valle de Guatemala, Ciudad de Guatemala, Guatemala, **14** Universidad de La Costa, Barranquilla, Colombia

* nlopez17@cuc.edu.co

## Abstract

### Background

The impact of quarantine in older adults have been reported in several studies with contradictory results, reporting from negative effects to no significant outcomes or even beneficial consequences. Heterogeneity in aging plays a role in each region, the aim of this study is to analyze the impact of quarantine on health conditions (physical and mental) and lifestyle in older adults in five Centro American countries during COVID-19 pandemic.

### Method

In this cross-sectional study, n = 712 older adults 60 years and older from Mexico, Guatemala, El Salvador, Honduras and Costa Rica were assessed by telephone. Sociodemographic data, physical and mental health, lifestyle and quarantine conditions were asked previous informed consent.

### Results

In general, mean of days in quarantine at the moment of the study was 142 days (approximately four months and three weeks). In the analysis of the impact of the days in quarantine effects were found on the frequency of falls, functional ability in Activities of Daily Living (ADL), general cognitive function, memory, orientation, language, frequency of drinking

**Data Availability Statement:** This study is part of an international multicenter project, entitled

"Cognitive Telephone Screening Study of Latin American and Caribbean Elderly Adults (Ref. INV.140-02-003-15). Data cannot be shared publicly because it contains participants´ potentially sensitive information. However, data are available at the Ethics Committee of Universidad de la Costa upon reasonable request (contact e-mail: ngonzale22@cuc.edu.pe).

**Funding:** The author(s) received no specific funding for this work.

**Competing interests:** The authors have declared that no competing interests exist.

alcohol, having a balanced diet, and being active cognitively. Some differences were found between countries.

## Conclusions

Effects of quarantine on older adults in Centro America, requires attention of governments and healthcare to prevent long term morbidity and disability, and to promote healthy aging.

## Introduction

The COVID-19 pandemic has rapidly spread all over the world. Although people of all ages can be infected by the novel Coronavirus, older adults were considered at special risk, especially if they had comorbidities like diabetes, hypertension, obesity, heart and respiratory diseases, cancer, lupus, renal insufficiency, and other chronic conditions [1]. In order to reduce the frequency of social contact and decrease the spread of Covid-19, the most important recommendations were the voluntary social isolation, lockdown and quarantine.

The impact of quarantine in older adults have been reported in several studies in high-income countries with heterogeneous results. Some studies have indicated that older adults may be less negatively affected by mental health outcomes than other age groups [2], and effects decrease with increasing age [3]. In the United States and Canada, a cross-sectional study involving adults from 18 to 91 years, reported that older adults showed better emotional well-being and less negative reactivity during pandemic compared to younger adults despite similar level of perceived stress [4]. In Spain, only older adults who had suffered COVID showed higher levels of anxiety, irritation and fear compared to those who had not, in general, most older adults maintained healthy lifestyles, with few changes in family and interpersonal relations during lockdown [5]. Moreover, mental well-being of German older adults was reported as largely unaltered during lockdown, outcomes and prevalence of depression, anxiety, loneliness, and distress did not differed markedly from those reported by studies undertaken before the pandemic [6].

On the other hand, some other studies have reported negative effects related to the social distancing and quarantine. In a review including individuals from Asia, Europe and America, the main outcomes reported were anxiety, depression, poor sleep quality and physical inactivity [7]. Loneliness during quarantine has been reported as strong predictor of depressive symptoms, anxiety and post-traumatic stress disorder in Spain [8]; in Belgium, older adults reported decrease in activity level and sleep quality, depression was strongly related to reported declines in wellbeing and cognitive functioning [9]. Additionally, the exacerbation of ageism, the general physical deterioration of older adults, reduced quality of life, and difficulties accessing services have been also noted as impact of the COVID-19 pandemic on older adults [10].

Older adults represent a heterogeneous group, which could explain the contradictory results found in the literature, besides, high income and low-and-middle income countries face different realities related to economy, population density, migration, and income. Inequities and disparities in health are observed among Latino individuals, who bear a disproportionate burden of COVID-19 related outcomes [11]. In this sense, this study focusses on the effects of COVID-19 quarantine on older adults in Centro American countries, that have not yet been known. The aim of this study is to analyze the impact of quarantine on health conditions (physical and mental) and lifestyle in older adults in five Centro American countries (Mexico, Guatemala, El Salvador, Honduras y Costa Rica) during COVID-19 pandemic.

## Materials and methods

### Study design and settings

This study employed a cross-sectional design, was conducted from June to October 2020 in community-dwelling older people from Centro American countries Mexico, Guatemala, El Salvador, Honduras and Costa Rica.

This study is part of the international multicenter project, entitled "Cognitive Telephone Screening Study of Latin American and Caribbean Elderly Adults (Ref. INV.140-02-003-15). Verbal and written consent was recorded, using digital devices or face-to-face in a deferred manner. The research was reviewed and approved by the Ethics Committee of the coordinate institution of the project, The Universidad de la Costa (Act No. 080).

### Participants

Participants were eligible if they were 60 years and older, community-dwelling, and were able to respond. Convenience samples were recruited, participants were assessed by telephone. Total sample consisted in n = 712 older adults, Mexico n = 200, Guatemala n = 200, El Salvador n = 149, Honduras n = 77, Costa Rica n = 86. Socio-demographic data are detailed in Table 1. In general, mean age of participants was 70.3 years old. Participants from Costa Rica were older and had less level of education than their counterparts. Most participants were women in all the countries, mostly married (except in Costa Rica where were mostly widow/er), and lived accompanied, either by their couple, spouse or family.

### Measurements and variables

For this study sociodemographic variables included: age ±years), sex ±female or male), marital status, education ±years), education level, living arrangement and housing.

Quarantine measures included the number of days at the moment of the interview.

For physical health measurements, individuals were asked if they currently had osteoarthritis, arthritis, osteoporosis, thyroid disease, hypertension, diabetes, dyslipidemia, stroke, Parkinson's, vascular dementia, Alzheimer's, and psychiatric disease; number of diseases resulted of the sum of those reported. Participants were also asked for the number of medicaments they consume, if they had fallen, and had suffered a fracture in the last year ±never, rarely, sometimes, often). Polypharmacy was considered if consuming more than four different medicaments every day [12]. Functionality in activities of daily living ±ADL) were measured by the Alzheimer´s Disease 8 ±AD8) [13].

Regarding mental health, were included a measure of general cognitive function by the Abbreviated version of the Montreal Cognitive Assessment ±MoCA) [14], which includes subscales for immediate and delayed memory, orientation ±spatial and temporal), and language, cut-off point <12 for mild cognitive impairment ±MCI) [15]. Depressive symptoms were assessed by the Geriatric Depression Scale [16], 5-item version [17, 18]. Additionally, participants were asked if they had been previously diagnosed with depression ±yes or no).

For lifestyle variables, participants were asked how frequently smoked, drank alcohol, had a balanced diet, practiced activities cognitively stimulating ±e.g. playing chess, sudokus, crosswords, reading), and the frequency they practiced physical activity. Response options were never, rarely, sometimes, often.

### Data analyses

Data were analyzed using Statistical Package for social Science ±SPSS) for Mac Version 24.0 ±IMB Corp.). The distribution of studied variables was explored using descriptive analysis.

**Table 1. Socio-demographic data of participants.**

| Variable | Total n = 712% (n) | Mexico n = 200% (n) | Guatemala n = 200% (n) | El Salvador n = 149% (n) | Honduras n = 77% (n) | Costa Rica n = 86% (n) |
|---|---|---|---|---|---|---|
| Age (years), Mean ±SD[a]*** | 70.3±8.2 | 70.2±8.2 | 69.2±6.9 | 68.3±7.1 | 67.8±8.0 | 78.6±8.5 |
| Age [b] *** | | | | | | |
| 60–64 | 30.8 ±219) | 34.0 ±68) | 30.0 ±60) | 36.2 ±54) | 41.6 ±32) | 5.8 ±5) |
| 65–69 | 24.0 ±171) | 22.5 ±45) | 26.0 ±52) | 28.2 ±42) | 27.3 ±21) | 12.8 ±11) |
| 70–74 | 16.9 ±120) | 13.5 ±27) | 23.5 ±47) | 16.1 ±24) | 13.0 ±10) | 14.0 ±12) |
| 75–79 | 12.2 ±87) | 14.0 ±28) | 11.0 ±22) | 8.1 ±12) | 11.7 ±9) | 18.6 ±16) |
| 80–84 | 9.4 ±67) | 9.0 ±18) | 6.5 ±13) | 9.4 ±14) | 2.6 ±2) | 23.3 ±20) |
| 85–89 | 4.6 ±33) | 5.0 ±10) | 3.0 ±6) | 2.0 ±3) | 0.0 ±0) | 16.3 ±14) |
| 90 + | 2.1 ±15) | 2.0 ±4) | 0.0 ±0) | 0.0 ±0) | 3.9 ±3) | 9.3 ±8) |
| Gender | | | | | | |
| Female | 69.9 ±498) | 70.0 ±140) | 72.5 ±145) | 67.1 ±100) | 63.6 ±49) | 74.4 ±64) |
| Male | 30.1 ±214) | 30.0 ±60) | 27.5 ±55) | 32.9 ±49) | 36.4 ±28) | 25.6 ±22) |
| Marital status [b] *** | | | | | | |
| Single / Widowed/Separated | 51.5 ±367) | 45.5 ±91) | 57.5 ±115) | 43.9 ±64) | 41.6 ±32) | 75.6 ±65) |
| Married | 48.5 ±345) | 54.5 ±109) | 42.5 ±85) | 57.0 ±85) | 58.4 ±45) | 24.4 ±21) |
| Education ±years), Mean ±SD [a]*** | 10.4±5.9 | 10.6± 6.3 | 11.1±5.8 | 11.6±6.0 | 10.7±5.0 | 6.8±4.5 |
| Education, level [b] *** | | | | | | |
| Iliterate/incomplete elementary | 21.1 ±150) | 18.0 ±36) | 14.0 ±28) | 30.2 ±45) | 7.8 ±6) | 40.7 ±35) |
| Complete elementary / Middle | 31.5 ±224) | 32.5 ±65) | 36.0 ±72) | 14.1 ±21) | 50.6 ±39) | 31.4 ±27) |
| High school | 10.8 ±77) | 19.5 ±39) | 11.5 ±23) | 5.4 ±8) | 7.8 ±6) | 1.2 ±1) |
| Collage | 28.9 ±206) | 19.0 ±38) | 30.5 ±61) | 39.6 ±59) | 32.5 ±25) | 26.7 ±23) |
| Postgraduate | 7.7 ±55) | 11.0 ±22) | 8.0 ±16) | 11.0 ±22) | 1.3 ±1) | 0.0 ±0) |
| Living arrangement [b]** | | | | | | |
| Alone | 12.6 ±90) | 9.5 ±19) | 13.5 ±27) | 14.1 ±21) | 0.0 ±0) | 26.7 ±23) |
| Couple/spouse | 38.8 ±276) | 45.5 ±91) | 33.5 ±67) | 45.0 ±67) | 44.2 ±34) | 19.8 ±17) |
| Family or roomie | 30.2 ±215) | 31.5 ±63) | 35.5 ±71) | 20.8 ±31) | 22.1 ±17) | 38.4 ±33) |
| Caregiver non-Familiar | 1.5 ±11) | 0.0 ±0) | 1.5 ±3) | 2.7 ±4) | 3.9 ±3) | 1.2 ±1) |
| Group of people non-familiar | 16.6 ±118) | 13.0 ±26) | 16.0 ±32) | 17.4 ±26) | 29.9 ±23) | 12.8 ±11) |
| Not specified | 0.3 ±2) | 0.5 ±1) | 0.0 ±0) | 0.0 ±0) | 0.0 ±0) | 1.2 ±1) |

Notes

*** = p < .000, SD = standard deviation

a = anova test

b = chi-square test.

Analysis of variance ±Anova) and Chi-square test were used appropriately for continuous and categorical variables. Pearson correlation was calculated to relate days in quarantine with age and education.

To explore the effect of the number of days in quarantine and the effect of the level of accomplishment of the lockdown, Multivariate Analyses of Variance ±MANOVA) was performed. Dependent variables were integrated all together if they correspond to "physical health", "mental health" and "lifestyle". Bonferroni adjusted alpha level of .017 was used, partial eta squared ±$\eta^2$) values were obtained to estimate the effect size, values larger than .14 ±or 14%) were considered a large effect.

**Table 2. Quarantine, health and lifestyle conditions of older persons in Centro American countries during pandemic due to COVID-19.**

| Variable | All countries n = 712% ±n) | Mexico n = 200% ±n) | Guatemala n = 200% ±n) | El Salvador n = 149% ±n) | Honduras n = 77% ±n) | Costa Rica n = 86% ±n) |
|---|---|---|---|---|---|---|
| Days in quarantine [a] *** | | | | | | |
| Mean ±SD | 142.1±33.7 | 145.8±34.4 | 146.5±33.7 | 120.3±43.1 | 143.6±60.4 | 166.3±25.6 |
| **Physical health** | | | | | | |
| Number of Diseases, Mean ±SD [a] *** | 1.71±1.44 | 1.81±1.50 | 1.59±1.45 | 1.41±1.16 | 1.61±1.38 | 2.38±1.56 |
| Hypertension ±yes) [b] *** | 50.0 ±356) | 47.0 ±94) | 39.0 ±78) | 53.0 ±79) | 55.8 ±43) | 72.1 ±62) |
| Diabetes ±yes) [b] * | 24.2 ±172) | 24.0 ±48) | 23.5 ±47) | 19.5 ±29) | 19.5 ±15) | 38.4 ±33) |
| Osteoporosis ±yes) | 11.8 ±84) | 15 ±30) | 10.5 ±21) | 10.7 ±16) | 7.8 ±6) | 12.8 ±11) |
| Dyslipidemia ±yes) [b] ** | 21.2 ±151) | 24.0 ±48) | 23.5 ±47) | 14.1 ±21) | 11.7 ±9) | 30.2 ±26) |
| Thyroid disease ±yes) | 11.1 ±79) | 11.0 ±22) | 9.0 ±18) | 9.4 ±14) | 11.7 ±9) | 18.6 ±16) |
| Medicaments, Mean ±SD [a] *** | 2.72±2.67 | 2.79±2.56 | 2.39±2.19 | 2.01±1.73 | 2.09±2.11 | 5.09±4.10 |
| Polypharmacy ±yes) [b] *** | 29.5 ±210) | 29.0 ±58) | 27.0 ±54) | 18.8 ±28) | 22.1 ±17) | 61.6 ±53) |
| Have fallen in the last year ±yes) [b] * | 53.7 ±382) | 53.0 ±106) | 58.5 ±117) | 55.7 ±83) | 37.7 ±29) | 54.7 ±47) |
| Have suffered a fracture in the last year ±yes) [b] * | 29.4 ±209) | 24.5 ±49) | 34.0 ±68) | 23.5 ±35) | 28.6 ±22) | 40.7 ±35) |
| Functionality ADL [b] *** | 42.0 ±299) | 38.5 ±77) | 30.5 ±61) | 40.3 ±60) | 76.6 ±59) | 48.8 ±42) |
| **Mental health** | | | | | | |
| MCI current ±yes) [b] *** | 19.2 ±137) | 14.5 ±29) | 15.5 ±31) | 16.8 ±25) | 20.8 ±16) | 41.9 ±36) |
| Previous depression [b] *** | 10.1 ±72) | 17.5 ±35) | 7.5 ±15) | 2.7 ±4) | 6.5 ±5) | 15.1 ±13) |
| Depressive symptom [b] *** | 32.7 ±233) | 38.0 ±76) | 26.0 ±52) | 21.5 ±32) | 45.5 ±35) | 44.2 ±38) |
| **Lifestyle** | | | | | | |
| Smoking ±yes) [b] ** | 8.0 ±57) | 11.0 ±22) | 9.8 ±18) | 2.7 ±4) | 14.3 ±11) | 2.3 ±2) |
| Drinking alcohol ±yes) [b] ** | 32.9 ±234) | 31.0 ±62) | 41.5 ±83) | 33.6 ±50) | 32.5 ±25) | 16.3 ±14) |
| Balanced diet ±yes) [b] *** | 65.3 ±465) | 94.0 ±188) | 94.0 ±188) | 33.6 ±50) | 32.5 ±25) | 16.3 ±14) |
| Cognitively active±yes) [b]*** | 74.7 ±532) | 74.0 ±148) | 81.5 ±163) | 63.8 ±95) | 61.0 ±47) | 91.9 ±79) |
| Physically active ±yes) [b] ** | 70.5 ±502) | 67.0 ±134) | 74.0 ±148) | 72.5 ±108) | 55.8 ±43) | 80.2 ±69) |

Notes

*** = p < .000

** = p < .01

* = p < .05, SD = standard deviation

a = anova test

b = chi-square test, ADL = Activities of daily living, MCI = Mild cognitive impairment.

## Results

Quarantine conditions are shown in Table 2. In general, mean of days in quarantine at the moment of the study was 142 days ±approximately four months and three weeks), Costa Rican older adults reported almost 20 days more. The number of days in quarantine was not associated to age and was negative associated to the education ±r = -.294, p = .001).

Countries differed in health and lifestyle conditions. Most reported diseases in all countries were hypertension and diabetes. Medicaments consumed ranged from 2.01 in El Salvador to 5.09 in Costa Rica. More than 50% reported they have fallen in the last year, except in Honduras ±37.7%). In general, 42% were independent in ADL and MCI was reported in 19.2% of participants from the five countries. Depressive symptoms were reported in 32.7% of participants in general. Regarding lifestyle conditions, in general 8% reported they smoked ±although in Costa Rica was only 2.3%), 32.9% drank alcohol reportedly, and the majority in all countries reported being cognitive and physically active.

**Table 3. Effect size of days in quarantine on physical health, mental health and lifestyle in older adults from Centro American countries during pandemic due to COVID-19.**

| Variable | All countries n = 712 F, $\eta^2$ | Mexico n = 200 F, $\eta^2$ | Guatemala n = 200 F, $\eta^2$ | El Salvador n = 149 F, $\eta^2$ | Honduras n = 77 F, $\eta^2$ | Costa Rica n = 86 F, $\eta^2$ |
|---|---|---|---|---|---|---|
| **Physical health** | **1.3, .21**[***] | **1.2, .25**[*] | **1.1, .36** | **1.3, .38**[**] | **4.4, .73**[***] | **1.2, .11** |
| Diseases ±number) | 1.2, .19 | 0.8, .19 | 0.8, .30 | 1.2, .36 | 3.0, .63[**] | 1.4, .13 |
| Medications ±number) | 1.1, .18 | 0.9, .20 | 1.0, .34 | 1.2, .36 | 2.0, .54 | 1.8, .16 |
| Have fallen during the last year ±freq.) | 1.5, .23[**] | 1.3, .27 | 1.9, .48[**] | 0.6, .21 | 5.1, .74[***] | 1.0, .09 |
| Have suffered fractures during the last year ±freq.) | 1.4, .22 | 1.4, .27 | 1.9, .48 | 1.1, .34 | 4.2, .71[**] | 0.6, .06 |
| Functionality in ADL | 1.6, .25[***] | 1.7, .32[**] | 1.1, .36 | 1.9, .46[**] | 16.9, .90[***] | 0.9, .09 |
| **Mental health** | **1.4, .23**[***] | **1.5, .30**[***] | **1.5, .44**[***] | **0.9, .32** | **3.5, .69**[***] | **1.4, .18**[**] |
| General cognitive function | 2.2, .31[***] | 1.8, .34[**] | 1.1, .35 | 1.7, .45[**] | 4.7, .73[***] | 1.7, .15 |
| Immediate memory | 2.0, .29[***] | 2.0, .36[**] | 1.1, .37 | 1.0, .32 | 4.5, .72[***] | 2.0, .17 |
| Delayed memory | 1.9, .28[***] | 1.6, .30 | 1.2, .37 | 1.3, .38 | 2.6, .60[**] | 2.0, .17 |
| Orientation | 2.1, .29[***] | 2.0, .36[**] | 1.2, .38 | 1.2, .36 | 57.6, .97[***] | 1.2, .11 |
| Language | 1.6, .24[**] | 1.2, .25 | 1.8, .47[**] | 1.0, .33 | 2.5, .59[**] | 2.9, .23[**] |
| Depressive symptoms | 1.6, .25[***] | 1.94, .34[**] | 1.8, .47[**] | .46, .17 | 2.5, .59 [**] | 1.9, .16 |
| **Lifestyle** | **1.8, .26**[***] | **1.7, .32**[***] | **1.3, .40**[**] | **1.2, .36** | **1.7, .52**[**] | **0.8, .08** |
| Smoking | 0.8, .14 | 0.6, .15 | 0.9, .31 | 0.8, .28 | 0.8, .33 | 0.5, .05 |
| Drinking alcohol | 2.2, .30[***] | 2.4, .39[***] | 1.5, .43[*] | 1.3, .38 | 5.0, .74[***] | 0.9, .08 |
| Having a balanced diet | 2.4, .32[***] | 1.5, .30 | 1.3, .40 | 1.3, .38 | 5.0, .74[***] | 0.9, .08 |
| Being active cognitively | 2.1, .30[***] | 2.3, .39[***] | 1.4, .42 | 1.5, .41 | 2.4, .58 | 1.9, .16 |
| Doing physical activity | 1.4, .22[**] | 1.5, .30 | 1.7, .46[**] | 0.9, .30 | 1.2, .42 | 0.2, .02 |

Notes

$\eta^2$ = Partial Eta Squared ±effect size), ±freq.) = frequency, ADL = Activities of daily living

** = $p < .017$ ±Bonferroni adjusted alpha)

*** = $p < .000$

In the analysis of the impact of the days in quarantine on physical health, mental health and lifestyle results are shown in Table 3.

Considering the five Centro American countries, days in quarantine had a significant large effect on the combined dependent variable physical health ±F = 1.3; p = .000; Wilkis´Lambda = .29) explaining 21% of the variance ±partial $\eta^2$ = .21). Specifically, days in quarantine contribute for differences in the frequency of older adults have fallen ±F = 1.5, p = .002, partial $\eta^2$ = .23) and the functionality in activities of daily living ±F = 1.6, p = .010, partial $\eta^2$ = .22). Effect sizes were large. Days in quarantine was not related to the number of diseases, the number of medicaments consumed, and the frequency of fractures during last year.

In addition, days in quarantine had a significant effect on the combined variable mental health, ±F = 1.44; p = .000; Wilkis´Lambda = .16) explaining 23% of the variance. Analyses of each individual dependent variable showed a significant contribution of days in quarantine explaining 31% of the variance on the general cognitive functioning, 29% of immediate memory and 28% of delayed memory, 29% of variance in orientation, 24% of the language ability, and 22% of variance on depressive symptoms was also explained by the days in quarantine.

Finally, days in quarantine also had an effect explaining 26% of the variance on the combined variable of lifestyle ±F = 1.80; p = .000; Wilkis´Lambda = .21). Specifically, days in quarantine had significant effect on the frequency of consumption of alcohol ±$\eta^2$ = .30), the

balanced alimentation $\pm\eta^2$ = 32.), being active cognitively $\pm\eta^2$ = .30), and practice physical activity $\pm\eta^2$ = .22). There was no contribution on the frequency that participants smoked.

Regarding the analyses of each country, differences were found. For Costa Rican participants impact of days in quarantine was only related to the combined variable of mental health, and specifically quarantine explained 23% of the variance in language. Contrary to participants from Honduras, where effects of quarantine were observed in almost all variables of physical and mental health as well as lifestyle, except in the number of medicaments, smoking and doing physical activity. Effect sizes are considered large. Older adults from Mexico, El Salvador y Honduras reported significant effects of days in quarantine in their physical health, specifically in their functionality in ADL, effect sizes were large in these countries.

Older adults also reported varied mental health effects related to days in quarantine. General cognitive function was significantly related to days in quarantine in Mexico, El Salvador y Honduras, explaining 34%, 45% and 73% respectively. Large effects of quarantine also were observed on memory $\pm$immediate and delayed) in Mexico and Honduras. Days in quarantine had also significant effects on depressive symptoms in older adults from Mexico, Guatemala y Honduras.

In lifestyle variables, smoking was not related to days in quarantine, contrary to drinking alcohol, where quarantine explained 39% of the variance in Mexico, 43% in Guatemala and 74% in Honduras. On the variable being active cognitively was also observed a significant effect of the days in quarantine, explaining the variance in Mexico $\pm$39%), Guatemala $\pm$42%), El Salvador $\pm$41%) and Honduras $\pm$58%).

## Discussion

Our study shows that quarantine due to COVID-19 pandemic had a negative impact on physical and mental health as well as on lifestyle in older adults from five Centro American Countries. Specific effects were found on the frequency of falls, functional ability in ADL, general cognitive function, memory, orientation, language, frequency of drinking alcohol, having a balanced diet, and being active cognitively; living with greater vulnerability and higher risk to long term morbidities.

These results add evidence to the controversy about the effects of quarantine in older adults, living in developed countries compared to those in low and middle-income countries. In the former, well-being trended to remain stable or even increase in 2020 compared to previous years, more social distancing were related to higher well-being, however it was found that more worry about health and financial consequences during pandemic were related to lower well-being [19], which is a reality in the Centro American countries, where health care systems are more fragile and vulnerable [20]. In a recent study, it was reported that Latin-American older adults living in their countries of origin experienced more negative impact on well-being due to COVID-19 than those Latin-American older adults living in the United States during pandemic, explains underline health disparities, discrimination, social exclusion and loneliness, that deeply influence an individual´s health and interact with social determinants of health [21].

Despite the idea on pandemic literature in older adults, that one of the few positives is the possibility that they may experience more adaptive coping in a crisis than younger adults [22] and early findings suggest higher resilience to the mental health effects of COVID-19 in community-dwelling older adults, this experience seems to be different between countries. In Centro American context, many older adults do not have the resources required to deal with the stress of COVID-19, this may include material $\pm$eg, lack of access to smart technology), social $\pm$eg, few family members or friends), or cognitive or biological $\pm$eg, inability to engage in

physical exercise or participate in activities or routines) resources [2]. In addition, pandemic expectations influence the extent to which older adults experience stress, there was found an association between expected income decline and negative affect in older adults when COVID-19 was considered as a perceived stressor [23]. Likewise, we must consider the socio-sanitary challenges that older people had to face in Central America countries, where access to specialized medical services and health prevention programs are scarce or with unequal Access [24]. In years prior to the pandemic, a progressive higher prevalence of non-communicable diseases was observed. Cardiovascular diseases were the more prevalent and were associated with excessive consumption of alcohol, tobacco, sedentarism and unhealthy diet [25].

Quarantine increased insolation, and it is well known that social participation is a key component for health and quality of life in aging. A worldwide multicenter study reported that social participation and life satisfaction was negatively impacted by confinement due to COVID-19, revealing psychosocial strain and highly mitigated by the use of digital technology [26]. However, older adults tend to be excluded from digital services ±due to decide not to use internet, lack of devices and network connectivity or inexperience using technology) struggling with the double burden of social and digital exclusion [27].

The present study must acknowledge some limitations. The first one is that evaluation was delivered through telephone, using a self-report questionnaire, however this study was conducted during pandemic when there was not another option. Another limitation is that due to the sampling strategy and sample size, representativeness is not guaranteed, although the inclusion of older adults from five different countries includes diversity and heterogeneity in sampling. In addition, given the cross-sectional design of this study, no causal relationships can be derived from the results, therefore longitudinal data are needed to provide stronger evidence, although lockdown is partially finished, and we hope not to go back to a restrictive confinement as in the summer of 2020 when this study was carried on. Anyway, longitudinal follow-up is considered in the future. All these limitations should be taken into account when considering the generalizability of our results, as well as the strengths of this project, emphasizing that is the first multicenter investigation reporting the impact of quarantine on indicators of physical and mental health, and lifestyle in older adults in the Centro American Region.

At the date of this manuscript, the epidemiological scenario is still complicated in this Region, according to official data, in México 17.7% of the total cases are 60 years and older ±representing 460,378 older adults) [28], 11.98% of the infected in Guatemala are older adults ±38,260 persons) [29], 15.1% in El Salvador ±12,387 older adults) [30], 13.1% in Honduras ±36,261persons) [31], and 6.8% of the total cases of COVID-19 in Costa Rica are older adults [32]. In Mexico, Guatemala and El Salvador the average of infections had been increased since then [33]. For this reason, the adoption of quarantine measures began in March 2020 except in Mexico where toughened measures were applied since May. In pandemic early stages, the level of compliance was quite high, associated with the closure of non-essential activities, closure of borders and even the imposition of fines on those who did not comply with the quarantine measures. However, citizens began to experience "pandemic fatigue" because of restrictive measures and the need to contribute with economic income to their families [34].

The COVID-19 pandemic has emphasized the needs and vulnerabilities that older persons have when it comes to their right to health; seen the highest mortality rates in older people with co-morbidities and with functional decline; and exposed the fragility of health systems to support older adults and consider their unique needs [35].

The current sanitary emergency and quarantine has disproportionately affected older adults, the demographic and epidemiological transitions within the Region require us to change how we treat and respond to our older person´s needs, especially in emergency situations. Finally, long term effects of quarantine on older adults in Centro America, requires

attention of governments and healthcare to prevent morbidity and disability, and to promote healthy aging.

## Acknowledgments

We thank the associations of older adults, geriatric clinical centers and Alzheimer's associations in Central America that made this research possible.

## Author Contributions

**Conceptualization:** Neyda Ma. Mendoza-Ruvalcaba, Marcio Soto-Añari, Norman López.

**Data curation:** Neyda Ma. Mendoza-Ruvalcaba, Raúl Gutiérrez-Herrera, Cecilia López, Marcio Soto-Añari, Miguel Ramos-Henderson, Loida Camargo, Nicole Caldichoury, Jorge Herrera-Pino.

**Formal analysis:** Neyda Ma. Mendoza-Ruvalcaba, Marcio Soto-Añari, José Calizaya-López.

**Investigation:** Neyda Ma. Mendoza-Ruvalcaba, Raúl Gutiérrez-Herrera, Cecilia López, Heike Hesse, Miguel Ramos-Henderson, Juan-Carlos Cárdenas-Valverde, Loida Camargo, Nicole Caldichoury, José Calizaya-López, Cesar Castellanos, Claudia García, María F. Porto, Norman López.

**Methodology:** Raúl Gutiérrez-Herrera, Marcio Soto-Añari, Jorge Herrera-Pino, Norman López.

**Resources:** Heike Hesse.

**Supervision:** Jorge Herrera-Pino, Claudia García, María F. Porto.

**Validation:** Marcio Soto-Añari, Miguel Ramos-Henderson, Juan-Carlos Cárdenas-Valverde, Loida Camargo, Cesar Castellanos, Norman López.

**Visualization:** Heike Hesse, Juan-Carlos Cárdenas-Valverde.

**Writing – original draft:** Neyda Ma. Mendoza-Ruvalcaba.

**Writing – review & editing:** Raúl Gutiérrez-Herrera, Cecilia López, Heike Hesse, Marcio Soto-Añari, Miguel Ramos-Henderson, Juan-Carlos Cárdenas-Valverde, Loida Camargo, Nicole Caldichoury, Jorge Herrera-Pino, José Calizaya-López, Cesar Castellanos, Claudia García, María F. Porto, Norman López.

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
