## [Decision Letter · Decision Letter 0]

12 Nov 2021

PONE-D-21-29050Impact of quarantine due to COVID-19 Pandemic on health and lifestyle conditions in older adults from Centro American countriesPLOS ONE

Dear Dr. Soto-Añari,

Thank you for submitting your manuscript to PLOS ONE. After careful consideration, we feel that it has merit but does not fully meet PLOS ONE’s publication criteria as it currently stands. Therefore, we invite you to submit a revised version of the manuscript that addresses the points raised during the review process.

We look forward to receiving your revised manuscript.

Kind regards,

Sanjay Kumar Singh Patel, Ph.D.

Academic Editor

PLOS ONE

2. PLOS ONE does not copy edit accepted manuscripts (https://journals.plos.org/plosone/s/criteria-for-publication#loc-5). To that effect, please ensure that your submission is free of typos and grammatical errors.

*For this observational study, please avoid causal-sounding language (such as 'impact' or 'effect') when reporting associations.

Reviewers' comments:

Reviewer's Responses to Questions

**Comments to the Author**

1. Is the manuscript technically sound, and do the data support the conclusions?

Reviewer #1: Partly

Reviewer #2: Yes

2. Has the statistical analysis been performed appropriately and rigorously? 

Reviewer #1: Yes

Reviewer #2: Yes

3. Have the authors made all data underlying the findings in their manuscript fully available?

Reviewer #1: Yes

Reviewer #2: Yes

4. Is the manuscript presented in an intelligible fashion and written in standard English?

Reviewer #1: Yes

Reviewer #2: Yes

5. Review Comments to the Author

Reviewer #1: The authors report the “Impact of quarantine due to COVID-19 Pandemic on health and lifestyle conditions in older adults from Centro American countries. This study found that the mental and physical health in older people is negatively affected during the COVID-19 quarantine. The authors suggest attention of governments and healthcare to prevent morbidity and disability, and to promote healthy aging.

Minor comments:

1) As the authors rightly pointed out “The present study must acknowledge some limitations” and have acknowledged few limitations in their study. So, to make this study more general the author should increase the sample size. The presented sample set of 712 is too small in number. If the authors were unable to increase the sample size, then, they should acknowledge this limitation in their manuscript.

2) Considering the socio-economic conditions, it’s not clear to me how strict was the lockdown in the Centro American countries and how strongly the subjects adhered to these quarantine measures.

3) Also, I am curious if the authors have any data about the weight loss or weight gain during this time.

Reviewer #2: The manuscript by Mendoza-Ruvalcaba et al. “Impact of quarantine due to COVID-19 Pandemic on health and lifestyle conditions in older adults from Centro American countries” is interesting and noteworthy. Authors performed cross-sectional study, n=712 older adults 60 years and older from Mexico, Guatemala, El Salvador, Honduras and Costa Rica were assessed by telephone. In conclusions authors have suggested that the effects of quarantine on older adults in Centro America, requires attention of governments and healthcare to prevent long term morbidity and disability, and to promote healthy aging. Furthermore, this reviewer feels the manuscript requires minor revision before its publication.

Comments

1. The language of manuscript may be polished (minor).

2. The abbreviations should be cross validated in the manuscript (First define them fully followed by abbreviation).

3. Please provide few information’s about health and lifestyle conditions in older adults before COVID-19 Pandemic in discussion (minor).

4. The authors may additionally provide one Figure as summary, challenges, or prospect of the present study.

---

## [Author Response · Author response to Decision Letter 0]

21 Jan 2022

Response to reviewers

Dear reviewers we appreciated your comments to our paper, there were incredible valuable to improve the paper robustness and clarify our ideas about the topic.

Reviewer 1

Minor comments:

1) As the authors rightly pointed out “The present study must acknowledge some limitations” and have acknowledged few limitations in their study. So, to make this study more general the author should increase the sample size. The presented sample set of 712 is too small in number. If the authors were unable to increase the sample size, then, they should acknowledge this limitation in their manuscript.

We added the following text to clarify that limitation:

“Another limitation is that due to the sampling strategy and sample size, representativeness is not guaranteed, although the inclusion of older adults from five different countries includes diversity and heterogeneity in sampling”.

2) Considering the socio-economic conditions, it’s not clear to me how strict was the lockdown in the Centro American countries and how strongly the subjects adhered to these quarantine measures.

We added the following text to clarify your observation, regarding the pandemic restrictions:

For this reason, the adoption of quarantine measures began in March 2020 except in Mexico where toughened measures were applied since May. In pandemic early stages, the level of compliance was quite high, associated with the closure of non-essential activities, closure of borders and even the imposition of fines on those who did not comply with the quarantine measures. However, citizens began to experience "pandemic fatigue" because of restrictive measures and the need to contribute with economic income to their families (34)

3) Also, I am curious if the authors have any data about the weight loss or weight gain during this time.

Unfortunately, we do not have that data.

Reviewer 2

1. The language of manuscript may be polished (minor).

The manuscript was reviewed by native English speaker to assure the quality of language.

2. The abbreviations should be cross validated in the manuscript (First define them fully followed by abbreviation).

We have completed your suggestion and were highlighted in the manuscript.

3. Please provide few information’s about health and lifestyle conditions in older adults before COVID-19 Pandemic in discussion (minor).

We added the following text to tackle your observation:

In years prior to the pandemic, a progressive higher prevalence of non-communicable diseases was observed. Cardiovascular diseases were the more prevalent and were associated with excessive consumption of alcohol, tobacco, sedentarism and unhealthy diet (25).

4. The authors may additionally provide one Figure as summary, challenges, or prospect of the present study.

We consider that information in discussion is covering this observation, mostly about challenges and prospect of the study.

Thanks for your feedback

Best regards

The authors

---

## [Editor Report · Decision Letter 1]

3 Feb 2022

Impact of quarantine due to COVID-19 Pandemic on health and lifestyle conditions in older adults from Centro American countries

PONE-D-21-29050R1

Dear Dr. Soto-Añari,

We’re pleased to inform you that your manuscript has been judged scientifically suitable for publication and will be formally accepted for publication once it meets all outstanding technical requirements.

Kind regards,

Sanjay Kumar Singh Patel, Ph.D.

Academic Editor

PLOS ONE

---

## [Editor Report · Acceptance letter]

3 May 2022

PONE-D-21-29050R1 

Impact of quarantine due to COVID-19 Pandemic on health and lifestyle conditions in older adults from Centro American countries 

Dear Dr. Soto-Añari:

I'm pleased to inform you that your manuscript has been deemed suitable for publication in PLOS ONE. Congratulations! Your manuscript is now with our production department. 

Kind regards, 

on behalf of

Dr. Sanjay Kumar Singh Patel 

%CORR_ED_EDITOR_ROLE%

PLOS ONE